# Long-Term Cost-Effectiveness through the Dental-Health FRAMM Guideline for Caries Prevention

**DOI:** 10.3390/ijerph19041954

**Published:** 2022-02-10

**Authors:** Thomas Davidson, Eva-Karin Bergström, Magnus Husberg, Ulla Moberg Sköld

**Affiliations:** 1Department of Health, Medicine and Caring Sciences, Linköping University, SE-581 83 Linköping, Sweden; magnus.husberg@liu.se; 2Department of Cariology, Institute of Odontology, Sahlgrenska Academy, University of Gothenburg, SE-405 30 Gothenburg, Sweden; eva-karin.bergstrom@vgregion.se (E.-K.B.); ulla.mobergskold@gmail.com (U.M.S.)

**Keywords:** adolescents, approximal tooth surfaces, caries prevalence, cost-effectiveness, fluoride varnish, school-based

## Abstract

A guideline called FRAMM, which is an acronym in Swedish for the most important parts of this guideline, namely “fluoride”, “advice”, “arena”, “motivation” and “diet”, was implemented in 2008 in the Västra Götaland Region in Sweden. This guideline included fluoride varnish applications performed at school twice a year at six-monthly intervals for all 12- to 15-year-olds, together with lessons on oral health. The aim of this analysis was to estimate the long-term cost-effectiveness, using prognostic calculations, of the FRAMM Guideline for 12- to 15-year-olds, compared with routine care, until the participants were 23 years old. A cost-effectiveness analysis was performed from a health care perspective, based on four years of verified data and seven years of prognosis. Data from FRAMM were combined with cost data from price lists in Sweden. The cost-effectiveness was analyzed by relating the difference in costs to the difference in the number of approximal surfaces with fillings and/or dentin lesions (DFSa). The analysis shows that FRAMM was considered dominant compared to the controls in all alternative scenarios, hence costs were prognosed to be lowered and outcomes were prognosed to be improved. A dental health program like the FRAMM Guideline with fluoride varnish during the caries risk period from 12 to 15 years is predicted to be cost-effective in the longer perspective. To further study the actual long-term caries increment after a preventive dental health program would be of great interest to verify these results.

## 1. Introduction

In the 21st century, caries remains a global health problem since untreated caries in permanent teeth affects 2.5 billion people, although caries has decreased among children during the last fifty years [1]. The main reason for tooth loss is still caries, which results in an impaired function to chew and assimilate food, loss of quality of life, and lower social status. The expenses for caries restoration as adults generate high costs for individuals as well as for society [2]. Hence, caries preventive methods are highly relevant.

There is strong evidence of caries reduction because of fluoride exposure at all ages [3,4]. Fluoride varnish is supported for caries prevention by many reviews and meta-analyses for the primary and the permanent dentition [5,6,7,8,9]. An updated Cochrane review of 22 trials found that fluoride varnish, applied two to four times a year, was associated with a reduction of 37% in caries in primary teeth and 43% in permanent teeth of children and adolescents, compared to placebo or no treatment, and the effect was not influenced by caries level of the population or by exposure to other sources of fluoride [10]. However, there is a barrier between research and clinic, why an important issue is to implement the results of evidence-based research into clinical practice. Furthermore, it is of importance to identify the best value for money, which means that limited resources should be used in the most effective way to obtain the largest health-care gain. A recent systematic review found several cost-effectiveness evaluations of caries preventive interventions in the literature, but these target primarily children at high risk [11]. There is a paucity of cost-effectiveness studies of general caries-preventive programs.

A guideline called FRAMM, which is an acronym in Swedish of the most important parts of this guideline, namely “fluoride”, “advice”, “arena”, “motivation” and “diet”, was implemented in 2008 for all 1- to 15-year-olds in the Västra Götaland Region in Sweden. This guideline included fluoride varnish applications performed at school twice a year at six-monthly intervals for all 12- to 15-year-olds, together with lessons on oral health. The guideline has been evaluated within a trial based on its effectiveness in preventing caries [12] and its cost-effectiveness [13]. However, no clear conclusions about the implementation of the program could be drawn from those results as reduced mean caries rates also came with increased costs.

It is of great interest to also evaluate if the FRAMM Guideline for 12- to 15-year-olds with fluoride varnish applications every six months has any impact on dental health and costs after the cessation of the intervention, and whether this would lead to any reduction in costs. Up to now, there is no study published that has evaluated the long-term cost-effectiveness of a similar guideline. Nevertheless, there are some earlier prevention studies on school-based fluoride rinsing that show a remaining effect on caries prevalence and incidence after cessation of the rinsing programs, but at that time, there were no evaluations of costs [14,15,16]. Furthermore, several studies have analyzed the cost-effectiveness of long-term fluoride varnish [17,18,19] but with inconclusive results [11].

The aim of this analysis was to estimate the cost-effectiveness, using prognostic calculations, of the FRAMM Guideline for 12- to 15-year-olds, compared with routine care, until the participants were 23 years old. In Sweden, there is tax-subsidised dental care for all children and adolescents, normally up to the age of 20, but up to the age of 23 in the Västra Götaland Region. After this age, there is dental insurance for adults. 

## 2. Materials and Methods

### 2.1. The FRAMM Guideline

The FRAMM Guideline for dental health promotion was implemented by the public dental service in all schools for 12- to 15-year-olds in the Västra Götaland Region in 2008 and was evaluated as described in two earlier studies [12,13]. One of the main goals for this guideline was to work with caries prevention in the most efficient way which meant to meet all 12- to 15-year-olds in groups at school instead of individually at the dental clinics. All adolescents are offered brief information on oral health and supervised dental flossing and approximal applications of fluoride varnish (Duraphat^®^, 2.26% F, Thepenier Pharma Industrie S.A.S, Mortagne, France) every six months, performed by dental nurses. In addition, they receive two scheduled lessons about oral health and tobacco use. The FRAMM Guideline is offered to all schools in the region as the basic preventive program, but a small number of the public dental clinics, approximately 10%, have a slightly extended program. These are situated in geographical areas with lower socio-economic status and therefore a higher risk of caries. In these areas, the number of fluoride varnish applications should be four times a year, compared to two times a year in the basic program. These areas are mainly situated in the suburbs of Gothenburg.

The FRAMM Guideline is offered for free to all 12- to 15-year-olds in this region and regular dental check-ups at the public dental clinics are free of charge. 

### 2.2. Data Extraction

Caries data, consisting of the number of decayed and/or filled approximal tooth surfaces (DFSa), for 13,490 adolescents born in 1993 who did not take part in the FRAMM Guideline at school and 11,321 adolescents born in 1998 who followed this guideline from age 12 to 15 were previously extracted from dental records in the region [12]. Those adolescents who had not been examined at the age of 12 were excluded, as their baseline caries experience could not be verified correctly. Both cohorts had participants from different geographical areas and with different caries prevalence and different socio-economic backgrounds.

### 2.3. Caries Prognostic Scenarios

As the analysis aims to be useful in health policy and for decision-makers, prognostic calculations with scenario analyses are used. The data of the children’s caries status and dental check-ups are almost complete between the age of 12 and 15 years old. After that, those data are prognosed in this study. The prognosis between 15 and 23 years is done by linear regression using ordinary least square (OLS), assuming a linear progression of DFSa. As the baseline in caries differed slightly between the FRAMM and the controls, the starting point was equalized by taking the average numbers of DFSa at age 12 for both groups. For years 12–15, the real development of DFSa in each group was used.

Three different scenarios for the prognosis of the caries development between 15 and 23 years were used. In Scenario A, the control group prognosis was performed by OLS while the prognosis of FRAMM started at the 15-year level but then followed the development of the control group. In Scenario B, OLS for the prognosis of both groups was used. Hence, in this scenario, the difference in DFSa between the groups found at age 15 is growing. In Scenario C, the same assumptions as in Scenario A were used but with a 10 per cent diminishing annual risk of new DFSa. This diminishing annual risk rate is not known and 10 per cent was based on a professional guess within the study group.

### 2.4. Costs and Cost-Effectiveness

The Consolidated Health Economic Evaluation Reporting Standards (CHEERS) checklist was applied to guidance this paper [20]. The cost-effectiveness analysis in this study estimates the incremental cost-effectiveness ratio (ICER) of implementing the FRAMM Guideline compared with routine care. The comparator (routine care) is the cohort of adolescents born in 1993 who had no fluoride varnish program at school. The analysis is performed from a health care perspective for 11 years. The outcome measurements used in the analysis are approximal surfaces with fillings and/or dentin lesions, DFSa. The reason for choosing approximal surfaces is because caries on these surfaces stands for the most need of restoration as adults.

Costs were calculated in Swedish kronor (SEK) and converted to euros (€) using an exchange rate of SEK 1 = € 0.1 (27 October 2021). The cost of the FRAMM Guideline was based on information from the public dental service in the Västra Götaland Region and related to the reimbursement for the guideline in 2017. This was € 12.50 per individual and year (between 12 and 15 years of age) for the application of fluoride varnish and € 5.15 per individual for the information on oral health on each of two occasions (12 and 15 years). This totals € 60.3 per individual included in the FRAMM Guideline. The time spent on the application of fluoride varnish was approximately 1–2 min per individual and the time spent on information on oral health was approximately 30–40 min per occasion.

The cost of fillings was calculated according to the official current pricelist in the region, which is € 112.1 per filling. Costs of dental check-ups were based on the national reference price, which is € 66 per visit. All costs are expected to represent their opportunity value within a health care perspective and no further adjustments on the costs are made. Costs in the future were discounted by 3 per cent annually, in line with the recommendations in Sweden [21].

### 2.5. Statistics

Microsoft Excel (Microsoft Corporation, Redmond, Washington, WA, USA) was used for the calculations and prognosis of caries development. The effect of the selected measurements was calculated as the difference between the predicted caries increment in the control group and the experimental group at age 23. Ordinary least squares (OLS) were used for the prognosis. *p* < 0.05 was applied for statistical significance.

### 2.6. Ethics

An ethical review of the extraction of data from the dental records was performed and approved by the Regional Ethical Review Board in Gothenburg, Sweden (Dnr: 273-14), prior to the first study by Bergström et al. [12].

## 3. Results

The prognosis of caries development found by Scenario A, B and C are presented in Figure 1. Scenario A shows that the mean number of DFSa with FRAMM at the age of 23 years is 1.11 compared with 1.20 of the controls, total mean prevention of 0.09 DFSa per participant. Scenario B shows larger differences in DFSa: 0.85 with FRAMM and 1.20 with the control, total mean prevention of 0.35 DFSa. Finally, Scenario C shows that FRAMM leads to 0.79 DFSa and that the control leads to 0.88, total mean prevention of 0.09 DFSa per participant.

The cost calculations include costs related to the intervention, dental check-ups, and fillings of developed DFSa. The intervention FRAMM cost € 60.3 per individual (equal to € 15.1 annually as presented in Table 1). Dental check-ups cost € 497.8 in the FRAMM group and € 505.5 in the control group during these years, and fillings cost less in the FRAMM group than in the controls. Using Scenario A, the cost of fillings was € 101 lower for FRAMM. By using Scenario B, the saving was € 390, and with Scenario C the saving was only € 7. The total cost was lower with FRAMM compared to the controls within all the scenarios. All the calculated costs by using Scenario A are presented in Table 1.

The cost-effectiveness was analyzed by relating the difference in costs to the difference in DFSa. In all scenarios, FRAMM is considered dominant compared to the controls, hence costs are lowered and outcomes are improved, see Table 2. The incremental cost-effectiveness ratios are also presented in a cost-effectiveness plane, see Figure 2. In this figure, one can see that all the results lead to reduced costs and less caries with FRAMM. Varying the discount rate between 0 and 5 per cent did not change this finding.

## 4. Discussion

The results from this study predict positive long-term effects both on dental health and costs up to the age of 23 years after the FRAMM Guideline. If this calculated prediction is accurate, this means health gain and economic gain for the individuals and for the society. It has previously been shown that adolescents who took part in the FRAMM Guideline had a significantly lower caries incidence on their approximal surfaces from 12 years up to 15 years compared with the control group who did not take part in this program [12]. This effect happened regardless of the caries prevalence at baseline at 12 years of age on these surfaces. Therefore, FRAMM had the ability to prevent new approximal caries lesions to a great extent for adolescents at these caries-risk ages when many approximal surfaces are at risk of caries [13].

The present study simulated three different scenarios for caries increments, and either one of those scenarios could be likely to be true. Scenario A simulated that both groups follow a linear regression of the control group after the age of 15, which would be a possible development. Scenario B is a somewhat more optimistic simulation with a linear regression of both groups, providing the most positive incremental cost-effectiveness ratio in the long-term. As a balance, Scenario C simulated a linear but diminishing annual risk of new DFSa, given a somewhat lower result. Nevertheless, all described scenarios showed the possibility for this dental health program to be cost-effective long after the participant has left the program. This emphasizes the long-term importance of such a program for adolescents at caries-risk ages, in addition to the regular use of fluoride toothpaste and regular dental check-ups.

The preventive fraction was between 25 and 34 per cent in all groups, hence clarifying the importance of a population-based strategy instead of a high-risk strategy, which is in accordance with earlier studies [22]. Suggesting FRAMM only for the high-risk group would lead to a rise in cost per participant, and the drop-out rate would probably be high. The ages of 12–15 years represent a period during which many tooth surfaces are at risk for caries and these ages are also a period with a high intake of junk food and soft drinks. The epidemiological data show that more new caries lesions will develop in the low to medium caries-risk group of adolescents, as this group is much larger than the high caries-risk group supporting a population strategy instead of a high-risk strategy [22]. With the FRAMM Guideline, all adolescents could take part in this program at school, which is of great interest from a democratic point of view, it does not harm anyone, and there are positive aspects of taking part in the same program as one’s school friends [23]. There is a challenge transferring evidence-based preventive programs into clinical practice [24], which is why guidelines such as FRAMM play an important role. The follow-up of compliance is of great importance and the FRAMM Guideline showed that 99 per cent of the school grades from 6 to 9 regularly took part in the program [12].

The main strengths of our study are the rigorous clinical trial and the large dataset it provides. The main limitation may be the reliability of the prognostic calculations. Using prognostic calculations to assess the long-term cost-effectiveness is sometimes used in health technology assessment (HTA). The uncertainty it creates can either be handled by using probabilistic sensitivity analyses or by scenario analyses [25]. The large dataset used in this study (more than 10,000 participants in each group) led to the conclusion that the statistical uncertainty of the DFSa results was very low, and a probabilistic analysis based on this would not have provided much more information. However, as there is uncertainty of how the DFSa would progress after the age of 15 years, the uncertainty of the decision is better described using scenario analysis. Instead of using prognostic calculations, one could have used simulation modelling techniques, for example with the use of a Markov model. However, in this case, that would not have led to different results as it is the number of caries that drives the cost-effectiveness in the analysis. Another limitation was the use of a historical control group (five years earlier) as it potentially could have been other factors than the FRAMM intervention that explain the differences. It would have been of great interest to compare the results of the FRAMM Guideline with a control group in a matching area without population-based interventions in another region in Sweden. However, most regions in Sweden have some sort of population-based caries-prevention program for children and adolescents. Additional limitations could be the transferability to other countries and settings. The costs of the intervention may be regarded as low, but it is the consequence of treating many adolescents in a fairly short time when using the school as the arena. Furthermore, the program is performed by dental nurses, and not dentists or dental hygienists, which would have been more expensive. In Sweden, all children and adolescents up to the age of 20 are entitled to free, tax-subsidised dental care with a focus on the prevention of oral disease. This can enable similar prevention programs in other Swedish regions but be more challenging in countries with other conditions. Nevertheless, cost-effective preventive programs for children and adolescents are of great interest in a wide context.

It is important to not only consider the clinical effects of a new method but also its economic consequences. As resources for dental care are limited, it is of importance to use these effectively. In this study, as well as in the former evaluations [12,13], the FRAMM Guideline has shown lower rates of DFSa compared to the control group. However, it is only in these long-term calculations that this led to a dominant result of FRAMM (lower cost and improved outcome). When a shorter period was used for the calculation of cost-effectiveness, improved oral health also provided higher costs [13]. The cost-effectiveness was the greatest for those with a high risk of caries, and it may have been interpreted as a reason to only include children with high caries risk in the program. However, as the long-term calculations in the current study show less DFSa and lower costs in general, there is no need to select certain risk groups.

There are some studies that the results of this study can be compared with. A study from Germany analyzing fluoride varnish application twice yearly in 6- to 18-year-olds from different caries-risk groups showed no cost-effectiveness in low-risk individuals in a clinic setting [18]. The explanation of the contrary results with the present study is probably the setting, as FRAMM is implemented in schools, providing a much lower cost than a clinical setting. Another study has analyzed the cost-effectiveness of caries-prevention programs with increased professional fluoride application or non-operative caries treatment and prevention compared with standard care among six-year-old children in the Netherlands [19]. This study was also within a trial, leading to the long-term effects not being captured, and this may be the main reason why both interventions led to higher costs. In terms of the cost-effectiveness of fluoride varnish programs, a study of a community oral health program compared fluoride varnish with fissure sealants in caries-free six- to seven-year-olds in an area with a high caries-risk in Wales. No differences in caries-prevention effects were found, but fluoride varnish proved to be less expensive and probably more cost-effective in the longer perspective [17].

Good oral health is important for the general quality of life and for public health in general. Caries is a major global public health problem since untreated caries in permanent teeth is the most prevalent health condition and affects 35% of the global population [1]. It has major negative impacts on public health on all levels such as individuals, families, communities and society, and is closely related to socio-economic status and to quality of life. Inequalities in oral health mirror those in general health. The costs of treating caries impose large economic burdens on all these levels. Therefore, for public health, it is of the greatest interest to minimize the disease, and the FRAMM Guideline is an example of an integrated upstream and community-based approach to prevent caries among children and adolescents in the caries-risk ages from 12 to 15 years. Good habits should be established early to be maintained for a long time. The efforts made besides fluoride varnish applications are lectures on subjects such as lifestyle habits which include intake pattern of junk food and sugar, smoking and snuffing habits, and these efforts are made at school where the individuals are five days a week and where the dental staff are integrated with the teachers at school. The dominant alternative preventive approach in dentistry is the downstream high-risk approach, focusing only on those individuals at risk for disease or those who have already got caries, and the efforts trying to change their behaviours have failed to effectively reduce oral health inequalities and may indeed have increased the oral health equity gap [1,26]. The results in this study show that the positive effect on caries prevalence remained up to the age of 23 years, but uncertainty prevails if those positive effects will remain further up in adult ages.

## 5. Conclusions

The dental health program, FRAMM Guideline, for 12- to 15-year-olds, was estimated to be cost-effective in the longer perspective. Three different simulation scenarios, with slightly different effects in caries-increment from 15 to 23 years, all predicted a positive long-term effect on costs, giving an economic gain for the individuals and for the society. To further study the actual long-term caries increment after a preventive dental health program would be of great interest.

## Figures and Tables

**Figure 1 ijerph-19-01954-f001:**
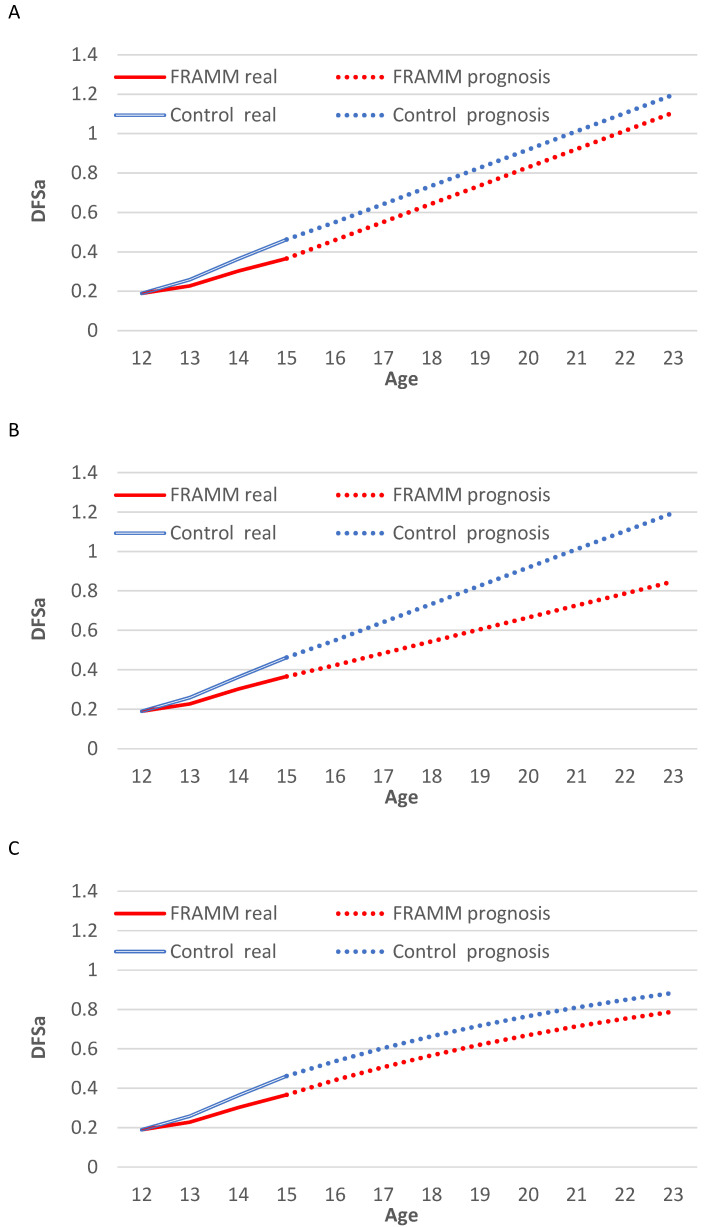
Illustration of the development of DFSa in the two studied groups using three different scenarios. (**A**) Scenario A: Both groups follow a linear regression of the control group after age 15; (**B**) Scenario B: Linear regression of both groups after age 15; (**C**) Scenario C: Same as in Scenario A but with diminishing annual risk (10%) of new DFSa.

**Figure 2 ijerph-19-01954-f002:**
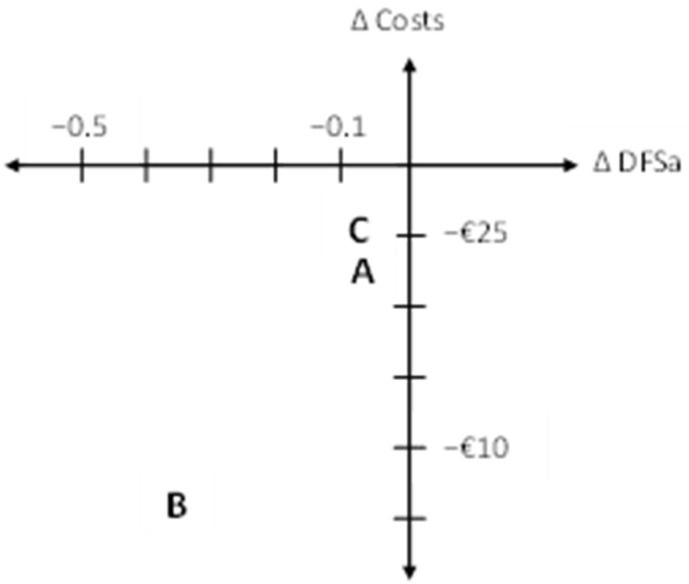
The incremental cost-effectiveness ratio (ICER) of scenarios A, B and C presented in a cost-effectiveness plane.

**Table 1 ijerph-19-01954-t001:** Costs (€) of FRAMM and the control groups using Scenario A.

Age	Intervention		Dental Checkups		Cost of Fillings		Total Costs		Total Costs Discounted	
	FRAMM	Control	FRAMM	Control	FRAMM	Control	FRAMM	Control	FRAMM	Control
12	15.1	0	66.0	66.0	21.2	21.2	102.3	87.2	102.3	87.2
13	15.1	0	37.5	33.8	25.4	29.0	78.0	62.8	75.8	61.0
14	15.1	0	39.6	48.4	33.9	40.7	88.6	89.1	83.5	83.9
15	15.1	0	37.9	40.5	41.1	51.8	94.0	92.3	86.0	84.5
16	0	0	39.6	39.6	51.4	61.5	91.0	101.1	80.9	89.9
17	0	0	39.6	39.6	61.8	71.9	101.4	111.5	87.4	96.2
18	0	0	39.6	39.6	72.1	82.2	111.7	121.8	93.6	102.0
19	0	0	39.6	39.6	82.5	92.6	122.1	132.2	99.2	107.5
20	0	0	39.6	39.6	92.8	102.9	132.4	142.5	104.5	112.5
21	0	0	39.6	39.6	103.2	113.3	142.8	152.9	109.4	117.2
22	0	0	39.6	39.6	113.5	123.6	153.1	163.2	113.9	121.5
23	0	0	39.6	39.6	123.9	134.0	163.5	173.6	118.1	125.4
Total	60.3	0	497.8	505.5	822.6	924.8	1380.8	1430.3	1154.6	1188.7

**Table 2 ijerph-19-01954-t002:** Cost-effectiveness of FRAMM compared with the controls using three different scenarios.

	Scenario A			Scenario B			Scenario C		
	FRAMM	Control	Diff.	FRAMM	Control	Diff.	FRAMM	Control	Diff.
DFSa	1.11	1.20	–0.09	0.85	1.20	–0.35	0.79	0.88	–0.09
Costs (€)	1154.6	1188.7	–34.1	1061.0	1180.3	–119.3	1060.1	1081.2	–21.1
ICER	Dominant			Dominant			Dominant		

DFSa = decayed and/or filled approximal tooth surfaces. ICER = Incremental cost-effectivenss ratio.

## Data Availability

All data can be acquired by the corresponding author of this article.

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
