# Peer review of "Long-Term Cost-Effectiveness through the Dental-Health FRAMM Guideline for Caries Prevention"

_ijerph, 2022, doi:10.3390/ijerph19041954_

Round 1

Reviewer 1 Report

Thank you for addressing my comments. It is a lot clearer, and the discussion section provides more of a discussion on the implications than previously. 

Author Response

Thank you.

Reviewer 2 Report

Many thanks for giving me the opportunity to review the manuscript entitled "Long-Term Cost-Effectiveness through the Dental-Health FRAMM Guideline for Caries Prevention".

This study estimated the long-term cost-effectiveness, using prognostic calculations, of the FRAMM Guideline (a programme which included fluoride varnish applications performed at school twice a year at six-monthly intervals), for 12- to 15-year-olds, compared with routine care, until the participants were 23 years old. Two groups (FRAMM group and routine) of adolescents were compared. The study concluded that the dental health program FRAMM Guideline for 12- to 15-year-olds was cost-effective in the longer perspective giving an economic gain for the individuals and for the society.

The article is written in a typical format. I did not notice any significant shortcomings with regard to the material presented in the article. Nevertheless, while reading the submitted manuscript, several questions arose and inaccuracies were noticed, which I recommend to fix not only personally for the reviewer but probably to readers too.

  1. It needs to be further explained how the schools / areas included in the FRAMM Guideline intervention group were selected. In the context of the FRAMM Guidelines, what does the information in lines 84-87 mean? It is worth providing a map showing the location of the intervention areas in the region.
  2. Table 1: Why do costs differ between Control and FRAMM at the age of 12-15 when calculating Dental checkups?
  3. There is a statement (lines 98-100): "Both cohorts had participants from different geographical areas and with different caries prevalence and different socio-economic background." Do these differences not need to be taken into account in procnosis calculations? It is clear that people in the higher socio-economic group receive better dental care services, eat healthier and therefore have better dental health.
  4. Lines 245-246: Correction is needed.

Thank you for considering my opinion. I encourage authors to keep on working to improve the manuscript.

Author Response

Comment 1: It needs to be further explained how schools/areas included in the FRAMM Guideline intervention group were selected. In the context of the FRAMM Guidelines, what does the information in lines 84-87 mean? It is worth providing a map showing the location of the intervention areas in the region?

Answer: Thank you. We realise it was not clear that all schools /areas are in the intervention group, and we have therefore explained this clearer in this section. We have also explained the small difference in number of fluoride applications in the extended program in areas with lower socio-economic status in the same section. We also explained that these areas are the suburbs of Gothenburg.

The idea of a map is interesting, but we think it anyway would be of limited use as most people will not be able to draw any conclusions from it. Only if you are very familiar with different areas in a region a map would provide a clear picture.

Added/revised text:

Line 77: “in all schools for 12- to 15-year-olds”

Line 84-89: “The FRAMM Guideline is offered to all schools in the region as the basic preventive program, but a small number of the public dental clinics, approximately 10%, have a slightly extended program. These are situated in geographical areas with a lower socio-economic status and therefore a higher risk of caries. In these areas the number of fluoride varnish applications should be four times a year, compared to two times a year in the basic program. These areas are mainly situated in the suburbs to Gothenburg.”

Comment 2: Why do costs differ between Control and FRAMM at the age of 12-15 when calculating Dental checkups?

Answer: As each dental checkup is given the same cost (€66 per visit), this difference is only related to the number of dental checkups, where the control group had slightly higher rates of dental checkups during these four years. We do not know the reason for this, but it could be that some of the children in the intervention group felt that they had already been checked in school, or that the slightly higher caries incidence in the control group caused this. However, we use the same rates (and costs) of dental checkups for both groups in the prognoses.

Comment 3: There is a statement (lines 98-100): Both cohorts had participants from different geographical areas and with different caries prevalence and different socio-economic background.” Do these differences not need to be taken into account in prognosis calculations? It is clear that people in the higher socio-economic group receive better dental care services, eat healthier and therefore have better dental health.

Answer: Thank you for this interesting question. The covariation between caries and socioeconomic factors is well known, but in this paper the effects were calculated on group levels. Therefore, the statement is important. The cohorts were not chosen due to socioeconomic status as the program was implemented in all schools in the region. A previous study has evaluated the program due to caries experience at 12-years of age, on an individual level, but this study aimed to estimate the cost-effectiveness of the whole program. In this region, as in Sweden in general, the dental care service is equal for all children and adolescents as this dental care is free of charge.   

Comment 4: Lines 245-246 Correction is needed.

Answer: Thank you, we have corrected this error.

Reviewer 3 Report

In my opinion an article that evaluates mainly the cost of different treatments it is not very relevant for the scientific field.

This article aimed to estimate the long term cost-effectiveness by using prognostic calculations.

What does this article bring new from the other article published by the same authors -Bergström EK, Davidson T, Moberg Sköld U. Cost-Effectiveness through the Dental-Health FRAMM Guideline for Caries Prevention among 12- to 15-Year-Olds in Sweden. Caries Res. 2019;53(3):339-346. doi: 10.1159/000495360. Epub 2019 Jan 16. PMID: 30650426.?

By whom was the Framm guideline implemented?

The authors cite two of their work and they mention that "However, no clear conclusions about implementation of the program could be drawn from those results as reduced mean caries rates also came with increased costs." could they please explain.

In my opinion this part is not relevant for the scientific world "Costs were calculated in Swedish kronor (SEK) and converted to euros (€) using an
exchange rate of SEK1=€0.1 (October 27, 2021). The cost of the FRAMM Guideline was based on information from the public dental service in the Västra Götaland Region and related to the reimbursement for the guideline in 2017. This was €12.50 per individual and year (between 12 and 15 years of age) for the application of fluoride varnish and €5.15 per individual for the information on oral health on each of two occasions (12 and 15 years).
This totals €60.3 per individual included in the FRAMM Guideline. The time spent on application of fluoride varnish was approximate 1-2 minutes per individual and the time spent on information on oral health was approximately 30-40 minutes per occasion.
The cost of fillings was calculated according to the official current pricelist in the region, which is €112.1 per filling. Costs of dental check-ups were based on the national reference price, which is €66 per visit. All costs are expected to represent their opportunity value within a health care perspective and no further adjustments on the costs are made. Costs in the future were discounted with 3 per cent annually in line with the recommendations in Sweden [22]."

The figures are not clear, they are blurry.

This part is not scientific- The Canadian doctor Sir William Osler coined the following expression already in the late 1800´s: “the health of the mouth is the window to the health of the body” which also with great clarity can be applied today more than hundred years later."

In the discussion chapter the authors have to compare their results with the result obtained from similar studies.

Author Response

Comment: What does this article bring new from the other article published by the same authors

Answer: As we write in lines 55-62, the information from the other two articles on the implementation of FRAMM are only partly helping a decision-maker to prioritize limited resources as those short term (within-trial) analyses showed improved outcome with additional costs. This article instead provides a longer time horizon, considering expected costs and outcomes for another seven years, resulting in ICERs that are clearly dominant, which hopefully will be beneficial for a decision-maker.

Comment: By whom was the FRAMM Guideline implemented?

Answer: Thank you for pointing this out. We can see that this information was missing and has added this to the sentence in lines 76: “by the public dental service”

Comment: The authors cite two of their work and they mention that “However, no clear conclusion about implementation of the program could be drawn form those results as reduced mean caries rates also came with increased costs”. Could they explain

Answer: The earlier studies have shown that the reduction in caries comes with additional costs. As we do not know the maximum societal threshold value for a reduction in caries, it is then not possible to say whether FRAMM is considered cost-effective or not. However, in this long-term study, the gain in caries reduction comes with decreased total cost, making the program clearly cost-effective.

Comment: In my opinion this part is not relevant for the scientific world…

Answer: This research belongs to the field of applied health economic and health care decision making. All this information is highly relevant within such a perspective. It is also recommended in the dominating checklist (CHEERS) for health economic studies.

Comment: The figures are not clear, they are blurry….

Answer: We agree that they have become a bit blurry, we think this happened when they were incorporated in the editing system. We have enclosed improved files.

Comment: This part is not scientific….. The Canadian doctor…

Answer: This part is now removed from the manuscript.

Comment: In the discussion chapter the authors have to compare their results with the results obtained from similar studies

Answer: We do this in lines 264-278. We couldn’t find other relevant studies to compare with.

Round 2

Reviewer 3 Report

The authors improved their article.

This manuscript is a resubmission of an earlier submission. The following is a list of the peer review reports and author responses from that submission.

Round 1

Reviewer 1 Report

ijerph-1466701

Long-Term Cost-Effectiveness through the Dental-Health  FRAMM Guideline for Caries Prevention

  1. From experience, DFS data is highly non-symmetrical in distribution. Therefore, it is not adequate to relate the initial value to a mean value. Non-parametric treatment of such data is a necessary condition for a reliable representation and interpretation of the data.
  2. The basis for the predictions of the condition at age 23 years is clinically collected DFSs between the ages of 12 and 15 years (real data). These collected data should be identical for the "FRAMM real" group for each of the three models used. The same holds for the "CONTROL real" group. However, Fig 1 shows considerable differences within both, the "FRAMM real" and the "Control real" group data. Thus, the prediction results as presented are invalid.
  3. To evaluate the results of the prognosis calculations, it is necessary to specify a measure of the variation in the data, e.g., the 95% confidence interval. This is the only way to interpret the data.
  4. In my experience, the EXCEL software package is not the most reliable that can be used to process predictive models. Fitting large amounts of data has not always worked reliably. I strongly encourage the authors to use a specialised professional program to analyse such large datasets.
  5. In 2.5. describe exactly which statistical procedures were used.

Author Response

Authors: Thank you for this thoroughly review. We have responded to all the questions below and have revised the paper substantially. Generally, our analysis aims to be useful in health policy and for decision-makers rather than stating the exact development of future caries development. That is the reason why we use prognostic modelling rather than a cohort simulation model. That is also the reason why we perform scenario analyses rather than probabilistic sensitivity analyses. This is now explained in the revised manuscript.

Reviewer: From experience, DFS data is highly non-symmetrical in distribution. Therefore, it is not adequate to relate the initial value to a mean value. Non-parametric treatment of such data is a necessary condition for a reliable representation and interpretation of the data.

Authors: Thank you. You are right that our data is highly non-symmetrical. However, we argue that the mean value anyway is the most relevant outcome, especially when it comes to costs as our dataset is very large. Even if using the mean as starting point is questionable, this does not affect the change in caries during the future years (or the costs).

Reviewer: The basis for the predictions of the condition at age 23 years is clinically collected DFSs between the ages of 12 and 15 years (real data). These collected data should be identical for the "FRAMM real" group for each of the three models used. The same holds for the "CONTROL real" group. However, Fig 1 shows considerable differences within both, the "FRAMM real" and the "Control real" group data. Thus, the prediction results as presented are invalid.

Authors: Those data are identical, it was just the figures (in former Fig 1) that were slightly different so that it looked like there were small differences. This has been improved in the revised version.

Reviewer: To evaluate the results of the prognosis calculations, it is necessary to specify a measure of the variation in the data, e.g., the 95% confidence interval. This is the only way to interpret the data.

Authors: Our confidence intervals of the mean values are very narrow due to the large sample (see our former analysis in Ref 14), so such variations do not provide decision makers with additional information. But there is instead uncertainty regarding what will happen with the caries development after the study trial, that is why we perform different scenarios. In the revised manuscript this is presented in both “Material and methods” and “Discussion”.

Reviewer: In my experience, the EXCEL software package is not the most reliable that can be used to process predictive models. Fitting large amounts of data has not always worked reliably. I strongly encourage the authors to use a specialised professional program to analyse such large datasets.

Authors: Thank you for this advice. However, for our purpose we think Excel is suitable and easy to review for many people.

Reviewer: In 2.5. describe exactly which statistical procedures were used.

Authors: We have added that ordinary least squares (OLS) were used.

Reviewer 2 Report

Thank you for your publication.  Please find my comments:

  • Acronyms should be avoided in the abstract (FRAMM, DFSa)
  • "No such guideline", in the abstract, needs to be expanded a little as this could be anything e.g., routine care, FV application yearly etc.
  • What does FRAMM stand for?
  • A previous CEA has been completed using the same population.  You need to justify and make it clear what is different about this piece of research, as it could be considered you are salami slicing data!
  • What about the implications of this guideline beyond 23? 
  • You're prognostic models are realistic, but why did you not consider use modelling approaches (e.g., cohort simulation model)?
  • In Model C, why 10% diminishing rate?
  • What perspective is your evaluation taking? I am assuming healthcare provider but this needs to be explicit
  • You have used the CHEERS checklist? A completed checklist attached as an appendix would be of benefit
  • I don't believe your treatment costs - only €12.50 for 2 applications of FV, but €5.15 for 30-40mins of oral health information. Have you considered fixed costs such as staff costs, overheads & capital costs which will be different between interventions.  Which filling material was used, was it an assumption it would be the same between the groups?  What about indirect restorations?
  • What does a DFSa of 1.2 & 1.11 mean - can you get 0.11 of a tooth surface affected? 
  • Page 4 line 147 - the word "about" should be removed. Exact costs should be reported in an economic evaluation
  • In table 2, the cost of the intervention is different to that presented in the text in Section 2.4
  • In table 2, have the costs be calculated each year using the same prognostic factor as the outcomes?  
  • In table 2, you appear to have only discounted costs from year 16, which is after the trial data, but the future costs would naturally be greater year on year as it is the opportunity cost of these future costs you are calculating 
  • You say the ICER is dominant, but you don't present the ICER.  Also, given the concerns mentioned about the cost data, this may not be the case.
  • The discussion doesn't really touch on the public health benefits and policy implications of FRAMM. 
  • Discussion lacks a strengths and limitations section.
  • Discussion does not mention decision modelling, which I feel, would have been more appropriate than caries prognostic models.  

Author Response

Authors: Thank you for this thoroughly review. We have responded to all the questions below and have revised the paper substantially. Generally, our analysis aims to be useful in health policy and for decision-makers rather than stating the exact development of future caries development. That is the reason why we use prognostic modelling rather than a cohort simulation model. That is also the reason why we perform scenario analyses rather than probabilistic sensitivity analyses. This is now explained in the revised manuscript.

Reviewer: Acronyms should be avoided in the abstract (FRAMM, DFSa)

Authors: Thank you, this is mostly changed. However, the acronym FRAMM is kept but explained the first time. This is because this acronym is the name of the intervention.

Reviewer: "No such guideline", in the abstract, needs to be expanded a little as this could be anything e.g., routine care, FV application yearly etc.

Authors: We have changed the wording to “routine care”

Reviewer: What does FRAMM stand for?

Authors: FRAMM is an acronym in Swedish of the most important parts of this guideline, namely “fluoride,” “advice,” “arena,” “motivation” and “diet”. This is explained in the revised manuscript.

Reviewer: A previous CEA has been completed using the same population.  You need to justify and make it clear what is different about this piece of research, as it could be considered you are salami slicing data!

Authors: Thank you. We mean that the following paragraph in the introduction section motivates the study:

“The guideline has been evaluated within trial based on its effectiveness in preventing caries [13] and its cost-effectiveness [14]. There is of greatest interest to also evaluate if FRAMM Guideline for 12-15-year-olds with fluoride varnish applications every six months has any impact on dental health and costs after cessation of the intervention. Up to now there is no study published that has evaluated the long-term cost-effectiveness of a similar guideline.”

Reviewer: What about the implications of this guideline beyond 23? 

Authors: Thank you for this question. In Sweden, there is a tax-subsidised dental care for all children and adolescents, normally up to the age of 20, but in the Västra Götaland Region up to the age of 23. After this age there is a dental insurance for adults. The Guideline FRAMM only includes children and adolescents in ages 3 to 16 years during periods of risk for caries. This information is added in the revised manuscript directly after presenting the aim.

Reviewer: You're prognostic models are realistic, but why did you not consider use modelling approaches (e.g., cohort simulation model)?

Authors: Thank you. Yes, we did consider using modelling approaches at first but thought that the prognostic modelling better suited our purpose. Generally, our analysis aims to be useful in health policy and for decision-makers rather than stating the exact development of future caries development. That is the reason why we use prognostic modelling rather than a cohort simulation model. That is also the reason why we perform scenario analyses rather than probabilistic sensitivity analyses.

Reviewer: In Model C, why 10% diminishing rate?

Authors: Actually, this figure in not known. Rather than being an exact value, this scenario analysis explores the outcomes of such diminishing rate.

Reviewer:  What perspective is your evaluation taking? I am assuming healthcare provider but this needs to be explicit

Authors: Thank you, you are right that we used a healthcare provider perspective (which was stated under the headline “Costs and cost-effectiveness”). In the revised manuscript we have also included this important information in the abstract.  

Reviewer: You have used the CHEERS checklist? A completed checklist attached as an appendix would be of benefit

Authors: We have enclosed a completed checklist to you but don’t think it is needed in the publication.  

Reviewer: I don't believe your treatment costs - only €12.50 for 2 applications of FV, but €5.15 for 30-40mins of oral health information. Have you considered fixed costs such as staff costs, overheads & capital costs which will be different between interventions.  Which filling material was used, was it an assumption it would be the same between the groups?  What about indirect restorations?

Authors: These values are low, we are aware of that. However, they are complete with overhead costs etc. The reason is that the applications of FV are provided at school which is very time effective. The same with the oral health information when many pupils get the information at the same time at school, hence the cost per participant gets very low. The costs of the intervention were explored in a former publication (see Ref 13 and 14).

Reviewer:  What does a DFSa of 1.2 & 1.11 mean - can you get 0.11 of a tooth surface affected? 

Authors: These are mean values (of a very large sample). A single participant can not have 0.11 DFSa.

Reviewer: Page 4 line 147 - the word "about" should be removed. Exact costs should be reported in an economic evaluation

Authors: Thank you, we have omitted “about” and instead written that it costed €497.8 in the FRAMM group and €505.5 in the control group.

Reviewer: In table 2, the cost of the intervention is different to that presented in the text in Section 2.4

Authors: Thank you. We understand that we need to be clearer about these values. In Table 2 the cost of the intervention is presented annually and the text in Section 2.4 was its total costs. But we have revised Section 2.4.  

Reviewer: In table 2, have the costs be calculated each year using the same prognostic factor as the outcomes?  

Authors: Apart from the intervention costs, the costs are directly related to the outcomes as DFSa lead to treatment costs.

Reviewer: In table 2, you appear to have only discounted costs from year 16, which is after the trial data, but the future costs would naturally be greater year on year as it is the opportunity cost of these future costs you are calculating 

Authors: We are discounting costs beginning on the second year (when the participants are 13).

Reviewer: You say the ICER is dominant, but you don't present the ICER.  Also, given the concerns mentioned about the cost data, this may not be the case.

Authors: Yes, the ICER is dominant. A dominant ICER should not be presented as a ratio as negative ratios are not possible to interpret (rather than being dominant).

Reviewer: The discussion doesn't really touch on the public health benefits and policy implications of FRAMM. 

Authors: Thank you for this comment which we agree with. We have added a few sentences in the discussion explaining the public health benefits from the Guideline FRAMM on dental health as well as on public health by introducing a population-based dental health programme during the most caries risk period for adolescents.

“Good oral health is important for the general quality of life and for public health in general. The Canadian doctor Sir William Osler coined the following expression already in the late 1800´s : “the health of the mouth is the window to the health of the body” which also with great clarity can be applied today more than hundred years later. Dental care for adults means large expenses for the individuals and for the society. Today in Sweden the costs for dental care are paid by 60% by the individuals and by 40% by the society and to minimize those expenses and to improve dental health among adults there is a need of a dental health programme during the ages with most tooth surfaces at risk for caries which is between the ages 12 and 16”.

Reviewer: Discussion lacks a strengths and limitations section.

Authors: Thank you. We did discuss this but not under a certain section. We have added the following sentence in the middle of the discussion section, just before we discuss prognostic modelling.

“The main strengths of our study are the rigorous clinical trial that and the large dataset it provided. The main limitations may be the transferability to other countries and settings, as well as the validity of the prognostic modelling.”

Reviewer: Discussion does not mention decision modelling, which I feel, would have been more appropriate than caries prognostic models.  

Authors: We are discussing pros and cons of our prognostic modelling method, but not specifically about models such as Markov or decision-trees. 

Round 2

Reviewer 2 Report

Thank you for addressing my comments and amending the manuscript  I still have a few issues with your revised manuscript, which are detailed below:

  • If your analysis aims to be useful in health policy and for decision-makers rather than stating the exact development of future caries development, then this needs to be much clearer. That you are only looking to evaluate to the end of the 23 year age group and NOT beyond.   I still think in your discussion you need to talk about future implications beyond the guidance as the outcomes thereafter are likely to be dissimilar given the intensive intervention the FRAMM group were given.
  • You to more clearly justify and state why prognostic modelling was used rather than a cohort simulation model.
  • If you don't know why 10%, then you need to say this
  • I am still not convinced about the costs. For FV, the price of the material, cost of staffing to complete etc. seem greater than €6.25/application.
  • Have you tested for normality to assume the data is normal, and therefore mean values can be used?
  • You have not answered my question: In table 2, have the costs be calculated each year using the same prognostic factor as the outcomes? 
  • Even if the ICER is dominant, a value would be beneficial and this has been demonstrated in other economic evaluations. At least you should have a cost-effectiveness plane demonstrating the result. 
  • Your discussion section remains weak. It still doesn't fully explore the public health benefits and policy implications of FRAMM specifically.  Nor does it have a meaningful strengths and limitations section.  The lack of discussion on alternative choice of models must be included